# Phytochemical Analysis and Anticancer Properties of *Drimia maritima* Bulb Extracts on Colorectal Cancer Cells

**DOI:** 10.3390/molecules28031215

**Published:** 2023-01-26

**Authors:** Khairallah Al-Abdallat, Maher Obeidat, Nidaa A. Ababneh, Suzan Zalloum, Sabal Al Hadidi, Yahya Al-Abdallat, Malek Zihlif, Abdalla Awidi

**Affiliations:** 1Bone Marrow Transplantation Unit, Jordan University Hospital, Amman 11942, Jordan; 2Hemostasis and Thrombosis Laboratory, School of Medicine, The University of Jordan, Amman 11942, Jordan; 3Cell Therapy Center, The University of Jordan, Amman 11942, Jordan; 4Department of Medical Laboratory Analysis, Faculty of Science, Al-Balqa Applied University, Al-Salt 19117, Jordan; 5Forensic Laboratory, Department of Public Security, Amman 11942, Jordan; 6Department of Pharmacology, School of Medicine, The University of Jordan, Amman 11942, Jordan; 7Department of Hematology and Oncology, Jordan University Hospital, Amman 11942, Jordan

**Keywords:** colorectal, *Drimia maritima*, anticancer

## Abstract

Cancer is a worldwide health problem and is the second leading cause of death after heart disease. Due to the high cost and severe side effects associated with chemotherapy treatments, natural products with anticancer therapeutic potential may play a promising role in anticancer therapy. The purpose of this study was to investigate the cytotoxic and apoptotic characteristics of the aqueous *Drimia maritima* bulb extract on Caco-2 and COLO-205 colorectal cancer cells. In order to reach such a purpose, the chemical composition was examined using the GC-MS method, and the selective antiproliferative effect was determined in colon cancer cell lines in normal gingival fibroblasts. The intracellular ROS, mitochondrial membrane potential, and gene expression changes in selected genes (*CASP8*, *TNF-α*, and *IL-6* genes) were assessed to determine the molecular mechanism of the antitumor effect of the extract. GC-MS results revealed the presence of fifty-seven compounds, and Proscillaridin A was the predominant secondary metabolite in the extract. The IC_50_ of *D*. *maritima* bulb extract on Caco-2, COLO-205, and the normal human gingival fibroblasts were obtained at 0.9 µg/mL, 2.3 µg/mL, and 13.1 µg/mL, respectively. The apoptotic effect assay indicated that the bulb extract induced apoptosis in both colon cancer cell lines. *D. maritima* bulb extract was only able to induce statistically significant ROS levels in COLO-205 cells in a dose-dependent manner. The mitochondrial membrane potential (MMP) revealed a significant decrease in the MMP of Caco-2 and COLO-205 to various concentrations of the bulb extract. At the molecular level, RT-qPCR was used to assess gene expression of *CASP8*, *TNF-α*, and *IL-6* genes in Caco-2 and COLO-205 cancer cells. The results showed that the expression of pro-inflammatory genes *TNF-α* and *IL-6* were upregulated. The apoptotic initiator gene *CASP8* was also upregulated in the Caco-2 cell line and did not reach significance in COLO-205 cells. These results lead to the conclusion that *D*. *maritima* extract induced cell death in both cell lines and may have the potential to be used in CRC therapy in the future.

## 1. Introduction

Cancer can be defined as a non-communicable disease with progressive uncontrolled cell growth/division with a probability to invade nearby and/or distant tissues and could appear in any part of the body [1,2,3,4]. Colorectal cancer, “CRC”, is the second most common cancer in both genders [5]. Natural products with anticancer therapeutic potential have earned an increasing interest in recent years. Pharmacological research of plant extracts as a source of secondary metabolites is probably the most important step in the identification of plant-derived compounds with therapeutic promise [6]. Plant extracts are a major source of phenolic compounds and other secondary metabolites, which have shown potential anticancer activities by regulating cell death pathways [7]. Furthermore, most of the anticancer drugs in the market are plant-derived such as taxanes (e.g., paclitaxel (Taxol) and docetaxel (Taxotere)) or synthetic compounds. Therefore, a need for basic and applied research to find novel cancer prevention and treatment options from plants as a source for safe and effective anticancer treatment [8].

To our knowledge, the cytotoxicity of the bulb extract of the medicinal plant *D. maritima* toward CRC cell lines was not previously investigated. Therefore, this research was initiated to study the therapeutic potential of *D*. *maritima* on CRC via assessment of the cytotoxic activity of *D. maritima* bulb extract on the colorectal Caco-2 and COLO-205 cancer cell lines. In this study, we prepared an aqueous extraction of the medicinal plant *D. maritima* bulb to assess the in vitro antiproliferative effects of *D*. *maritima* bulb extract against colorectal cancer cells. We also aimed to determine the mechanism of action of active *D*. *maritima* bulb extract against colorectal cancer cells. Finally, we aimed to examine the gene expression of some cell growth-related genes in bulb extract-treated colorectal cancer cells using quantitative RT-PCR.

## 2. Results

### 2.1. Identification of Chemical Constituents of D. maritima Bulb Extract by GC-MS 

Analysis of the bioactive constituents of the bulb extract of *D. maritima* by GC-MS demonstrated the presence of 57 compounds and their chromatogram, as shown in Figure 1 and Table 1. Major peaks determined in the GC-MS chromatogram and their corresponding components were recognized according to NIST 20 and Willy 19 Libraries. The bulb extract was found to include a variety of essential phytochemicals. The results of GC-MS analysis showed that *D. maritima* bulb extract was rich in the phytosterol “Campesterol” containing ~14%, 2,3-Butanediol (glycol and secondary alcohol) and 5-Hydroxymethylfurfural (furans, an arenecarbaldehyde, and primary alcohol) (~10% each), and ~8% for each of the fatty acid derivatives “Hexadecanoic acid ethyl ester” and “9-Octadecenamide (Z)-”. Interestingly, the cardiac glycoside “Proscillaridin” proved to be one of the predominant secondary metabolites in the bulb extract of *D. maritima* with about 5% content which may play an effective role as a pharmaceutical anticancer agent (Table 1).

### 2.2. Drimia maritima Bulb Extract and ProA Can Selectively Inhibit the Proliferation of COLO-205 and Caco-2 Cancer Cells

The MTT assay was performed to evaluate the cell viability and proliferation of COLO-205 and Caco-2 cells upon treatment with different concentrations of *D*. *maritima* bulb extract, ProA, and Doxorubicin (positive control). The treatment of COLO-205 and Caco-2 with different concentrations of *D*. *maritima* bulb extract showed cytotoxic effects proportional to the concentration of *D*. *maritima* bulb extract used. The IC_50_ of *D*. *maritima* bulb extract on COLO-205 cell lines was ~2.3 μg/mL and for Caco-2 cells was ~0.9 μg/mL (Figure 2). In order to determine whether *D*. *maritima* bulb extract has cytotoxicity effects on normal human cells, normal HGF cells were treated with the plant extract using the same concentrations used in colon cancer cell lines and incubated for 48 h. The results showed that *D*. *maritima* bulb extract has minimal cytotoxic effects on normal human cells at low concentrations, and the IC_50_ was achieved at a relatively high concentration of 13.1 μg/mL (Figure 2), indicating that *D*. *maritima* bulb extract potentially has selective cytotoxic effects on cancer cells but not on normal cells. From the reported IC50s, an estimation for the selectivity index can be found to be approximately 5.7 for the COLO-205 cells and 14.55 for Caco-2 cells.

On the other hand, to evaluate the cytotoxic effect of ProA, which is one of the major active compounds of *D*. *maritima* (Table 1), COLO-205 and Caco-2 cells, as well as the normal HGF cells, were treated with different concentrations of ProA ranging from 0.0024 μg/mL to 1.32 μg/mL. The results, as indicated in Figure 2B, showed a significant increase in Caco-2 and COLO-205 cell death, and the IC_50_ was achieved at 0.0029 μg/mL and 0.012 μg/mL, respectively. Moreover, ProA showed a minimal cytotoxic effect on normal HGF, and the IC_50_ was achieved at 0.229 μg/mL.

Doxorubicin was also used to assess the efficacy and the potential of ProA and bulb extract as natural cancer therapeutics. Importantly, ProA showed a more powerful cytotoxic effect than *D*. *maritima* bulb extract on both Caco-2 and COLO-205 cell lines when compared to Doxorubicin, which was achieved at IC_50_ at 0.78 μg/mL for COLO-205 and 0.59 μg/mL for Caco-2 (Figure 2C).

### 2.3. D. maritima Bulb Extract Can Induce Early and Late Apoptosis in Colon Cancer Cell Lines

In order to clarify whether *D*. *maritima* bulb extract can induce cell apoptosis in colon cancer cells, *D. maritima*-treated and -untreated Caco-2 and COLO-205 cells were stained with Annexin V and PI to analyze different apoptotic stages using flow cytometry. When compared to untreated cells (control cells), *D*. *maritima* bulb extract was able to significantly increase both the early and late cell apoptotic percentages in treated colon cancer cells in a concentration-dependent manner. In the COLO-205 cell culture, there was a statistically significant increase in the percentage of early and late apoptotic cells (Figure 3A). These results clearly emphasize that *D*. *maritima* bulb extract is able to induce cell early and late apoptosis in Caco-2 and COLO-205 cell lines in a concentration-dependent manner (Figure 3B).

### 2.4. D. maritima Bulb Extract Induces the Production of ROS in Colon Cancer Cells

ROS generation was measured after treatment with *D. maritima* bulb extracts in colon cancer cells. *D. maritima* bulb extract was only able to induce statistically significant ROS levels in COLO-205 cells in a dose-dependent manner when treated with concentrations above 2 µg/mL (i.e., 3.5, 2.75, and 2 µg/mL; *p*-value > 0.001) (Figure 4). Although *D. maritima* bulb extract was able to induce ROS in Caco-2 cells at lower plant extract concentrations (0.5 and 1 µg/mL), these results were statistically non-significant (Figure 4). Moreover, *D. maritima* bulb extract was not able to induce a statistically significant level of ROS in Caco-2 cells. Perhaps, this could be referred to the lower concentrations of *D. maritima* bulb extract used to treat Caco-2 cells (1.5, 1, and 0.75 µg/mL) when compared to that used to treat COLO-205.

### 2.5. D. maritima Bulb Extract Affects Mitochondrial Membrane Potential (ΔΨm) in Colon Cancer Cells

The effect of *D. maritima* bulb extract on mitochondrial membrane potential (ΔΨm) was assessed in colorectal cancer cells using TMRE cell-permeant red-orange dye to label active mitochondria. FCCP was used as a positive control, which can significantly diminish mitochondrial membrane potential. The results showed that treatment of COLO-205 cells concentrations of *D. maritima* bulb extracts above 2 µg/mL (2.75 and 3.5 µg/mL) resulted in a statistically significant (*p* < 0.001) decrease in ΔΨm of COLO-205 cells (4.569 ± 0.726, and 0.666 ± 0.72, respectively) when compared to control cells (Figure 5). Although the effect of *D. maritima* bulb extracts on ΔΨm in COLO-205 cells at 2 µg/mL concentration was minor, it was statistically significant (15.38 ± 0.72; *p* < 0.05). However, treatment of COLO-205 with 1 µg/mL *D. maritima* bulb extract showed no significant reduction in their ΔΨm.

On the other hand, treatment of Caco-2 cells with different concentrations of *D. maritima* bulb extract of 0.5, 0.75, 1, 1.5, and 2 µg/mL resulted in a statistically significant (*p* < 0.001) reduction in their ΔΨm (260 ± 194.4, 250.3 ± 194.4, 290.3 ± 194.4, 326.0 ± 194.4, and 287.7 ± 194.4, respectively) when compared to the control cells (Figure 5). These results confirmed that *D. maritima* bulb extract could significantly reduce ΔΨm in COLO-205 cells in a dose-dependent manner and also in the Caco-2 cell line.

### 2.6. The Impact of D. maritima Bulb Extract on Gene Expression in Colon Cancer Cell Lines

In order to figure out changes in colon cancer cell lines at the molecular level after the observed growth-inhibitory effect of *D. maritima* bulb extract, RT-qPCR was applied to assess gene expression of multiple genes: *Casp8*, *TNF-α*, and *IL-6* in COLO-205 and Caco-2 cells pre-treated with different concentrations of *D. maritima* bulb extract.

In COLO-205 cells, there was a non-significant upregulation in *Casp8* gene by when the cells were treated with the extract at a concentration of 3.5 μg/mL (Figure 6A). Moreover, treatment with the extract for 48 h at 2.75 and 3.5 μg/mL resulted in statistically significant elevations in the target gene *TNF-α* (5.59-fold and 4.8-fold, respectively). Although there was an elevation in the expression of the *TNF-α* gene when cells were treated with the extract at 2 μg/mL by 1.65-fold, this elevation was statistically non-significant (Figure 6B). Furthermore, there was a non-significant elevation in the expression of the pro-inflammatory cytokine gene *IL-6* when COLO-205 was treated at extract concentrations 2, 2.75, and 3.5 μg/mL (1.2-, 1.6-, and 2.1-fold, respectively) (Figure 6C).

In comparison to the control group (untreated cells), the pro-inflammatory *TNF-α* expression levels in Caco-2 cells were significantly increased by 6.5- and 6.4-fold following treatment with 2 and 1.5 μg/mL of *D. maritima* bulb extracted for 48 h, respectively (*p* < 0.0001), and 5.2-fold when cells were treated with 1 μg/mL (*p* < 0.0007) (Figure 6D). Although the expression levels of *TNF-α* elevated by 3.3- and 3.6-fold when Caco-2 cells were treated with 0.75 and 0.5 μg/mL, respectively, these elevations were not significant. The expression of the *Casp8* gene was significantly upregulated (1.5-fold) following 48 h treatment with 1.5 μg/mL (*p* < 0.001) in Caco-2 cells. Moreover, the expression still elevated significantly by 1.31- and 1.3-fold when decreasing the concentration to 1 and 0.75 μg/mL (*p* < 0.041 and <0.014), respectively (Figure 6E). An elevation also occurred when Caco-2 cells were treated with 0.5 μg/mL, yet, this elevation was statistically non-significant. Furthermore, results showed that *D. maritima* bulb extract could induce the activation of the effector *Casp8* gene in Caco-2 cells (Figure 6E). With respect to the expression of pro-inflammatory cytokine gene *IL-6*, results showed that there was a statistically significant upregulation (6.5-fold increase) when Caco-2 cells were treated with 1.5 μg/mL (*p* < 0.0244) when compared to control cells (Figure 6F). Although there was an increase in the expression of gene *IL-6* (3.4-, 3.2-, and 5.5-fold) upon treatment of Caco-2 cells with 0.5, 0.75, and 1 μg/mL of *D. maritima* bulb extract, respectively, these elevations were statistically non-significant.

In general, the results showed that *D. maritima* bulb extract could affect the expression of cell proliferation-related genes that were examined in this study in both cell lines, i.e., it can upregulate the expression of inflammatory cytokine genes *TNF-α* and *IL-6* and the apoptotic initiator gene *Casp8*.

## 3. Discussion

Plants are an interesting and cost-effective natural source of potential therapeutic candidates, and their secondary metabolites, phenols, and extracts have been extensively investigated for a mode of action on cancer cell death and their anticarcinogenic properties [9]. In the current study, the anticancer potential of a novel medicinal plant common in Mediterranean regions, namely *D. maritima* was investigated. *D. maritima* has long been used for the management and treatment of many pathological conditions [10,11,12,13,14]. In cancer settings, several preceding in vitro studies have reported potent anticancerous and selective cytotoxic effects of *D. maritima* on different cancer types, including breast, lung, lymphoma, and prostate cancer [10,11,13,15]. However, [16] the cytotoxic effect of *D*. *calcarata* bulb extracts against *p53* mutant HT-29 and *p53* wild-type Caco-2 colorectal cancer cells has recently been demonstrated. Herein, to our knowledge, the current study is the first to investigate the anticancerous effect of *D. maritima* bulb extracts on COLO-205 and Caco-2 colorectal cancer cell lines.

In order to evaluate the potential antitumor effect mediated by *D. maritima* bulb extract on the cell proliferation of colon COLO-205 and Caco-2 cancer cell lines, an MTT assay was used to measure mitochondrial activity in viable cells. Interestingly, *D. maritima* bulb extract showed a dose-dependent antiproliferative effect against both colon cancer cell lines (Caco-2 and COLO-205). *D. maritima* bulb extract was able to achieve IC_50_ at 0.92 and 2.3 µg/mL in Caco-2 and COLO-205, respectively. In this study, ProA, which accounts for about 5% of the *D. maritima* bulb extract, was one of the major active phytochemical compounds with potent anticancer activity on different cancer types, including, but not limited to, glioblastoma, lung, and prostate cancers [17,18]. Accordingly, it was assumed that the cytotoxic activity of *D. maritima* bulb extracts on colon COLO-205 and Caco-2 cell lines could be achieved, at least in part, through ProA as an active compound. Interestingly, ProA was highly effective in inhibiting COLO-205 and Caco-2 cell lines and achieved IC_50_ at concentrations of 0.0029 µg/mL and 0.012µg/mL, respectively. Additionally, both *D. maritima* bulb extract and ProA express minimal cytotoxic effects against normal HGF when compared to cancer cells (IC_50_ equals 13.1 µg/mL and 0.229 µg/mL, respectively), indicating that their cytotoxic activities are selective to colon cancer cells, with a very attractive selectivity index of 14.6 for Caco-2 and 5.6 for COLO-205. Interestingly, the obtained IC50 values in this study are relatively lower than the previously reported IC50 against different breast cancer cell lines. For example, Hamzeloo-Moghadam reported that the IC50 values for the non-hemolytic ethanol extract of U. maritima against breast cancer MCF7 cells were 11.01 µg/mL. Additionally, IC50 values for the non-hemolytic acetone extract of U. maritima against breast cancer MCF7 cells was 6.01 µg/mL, while the Doxorubicin drug (positive control) exhibited a cytotoxic effect with a higher IC50 value (52.35 µg/mL) [19]. Another example is the study conducted by Obeidat and Sharan, which found that the effective doses of *D. maritima* that inhibited 50% of growth (IC50) of MCF-7 and MDA-MB-468 cells were 20.48 ± 1.17 µg/mL and 25.74 ± 2.05 µg/mL, respectively. Interestingly, Obeidat and Sharab’s study revealed that *D. maritima* displays significantly lower cytotoxicity against AGO1522, a normal human fibroblast cell, with IC_50_ values of 43.5 ± 1.73 µg/mL [13].

Under normal physiological conditions, apoptosis “programmed cell death” is considered a mechanism to eliminate aged or abnormal cells; however, in cancer settings, cancer cells are, in general, resistant to apoptosis and induce apoptosis “therapeutically”, which is considered an effective way to eliminate cancer cells [20]. The findings of this study showed that when compared to untreated cells (control), *D. maritima* bulb extract was able to induce early and late cell apoptosis in a concentration-dependent manner.

*D. maritima* bulb extract was able to significantly (*p* < 0.0001) induce ROS production in both Caco-2 and COLO-205 cancer cell lines after 48 h of treatment in a dose-dependent manner. As aforementioned, intracellular ROS generation in large amounts is considered a mechanism to trigger cell apoptosis, in part, through triggering the endoplasmic reticulum stress [21], and thus, colon cancer cells apoptosis investigated in this study could be a result of excessive intracellular ROS generation mediated by *D. maritima* bulb extract. On the other hand, excessive ROS generation can cause cell cycle arrest as described previously [22], but still, our results did not confirm the finding; at the same time, this possibility cannot be excluded.

In order to further understand the mechanisms that promoted apoptosis in colon cancer cells upon treatment with *D. maritima* bulb extract, studying the mitochondrial membrane potential, ΔΨm, is of significant value in this context. In particular, ΔΨm has considered one of the key markers of apoptosis [23]. Accordingly, the changes in the ΔΨm were evaluated by tracking TMRE fluorescent signal in colon cancer cells (Caco-2 and COLO-205 cell lines) after treatment with different concentrations of *D. maritima* bulb extract for 48 h. Importantly, when compared to untreated cells (control), a significant reduction in ΔΨm was observed in colon cancer cells (*p* < 0.0001), and this effect was dose-dependent. Notably, the Colo-205 represents a metastatic colorectal cancer, as this cell was isolated from the ascitic fluid of a 70-year-old Caucasian male with carcinoma of the colon. The patient had been treated with 5-fluorouracil for 4–6 weeks before the removal of the fluid specimen. Additionally, Caco-2 cells represent relatively sensitive cells toward the 5-fluorouracil. Interestingly, the extract used in this study shows a relatively higher activity toward the colorectal Caco-2 cells. Interestingly, the apoptosis assay has also shown less apoptosis percentage than those seen in the case of the Caco-2 cells. These results inversely correlate with the ROS results, indicating that there is a negative correlation between ROS generation and ΔΨm, which is consistent with the results of [24], and showed that curcumin is able to induce apoptosis in melanoma cancer cells by increasing the production of ROS and reducing ΔΨm

Finally, in order to track the molecular pathway through which *D. maritima* bulb extract was able to induce apoptosis in colon cancer cell lines, RT-qPCR was used to quantify the following target genes: genes encoding for pro-inflammatory cytokines, including TNF-α and IL-6, as well as the gene encoding for caspase-8. TNF-α is an important pleiotropic pro-inflammatory cytokine that, once produced at the site of inflammation, can mediate a wide range of cellular responses that include but are not limited to stimulating the production of pro-inflammatory cytokines, affecting the survival and proliferation of cells and/or inducing cell death under certain circumstances [25]. In fact, TNF-α can positively regulate the survival and proliferation of cells if it activates certain NF-kB-dependent genes involved in cell survival and proliferation [26,27]. The latter can be achieved through the activation of distinct caspase-8 activation pathways [27]. Furthermore, it has been reported that upregulating the gene expression of *TNF-α* and *IL-6* can participate in inducing Caco-cell death [28]. This explains why the above-mentioned three gene targets are examined in this study. Importantly, the results showed that upon treatment of both colon cancer cell lines with the *D. maritima* bulb extract, there was a statistically significant increase in the expression of all three gene targets tested in this study. This may indicate that *D. maritima* bulb extract induces apoptosis in colon cancer cells through the *TNF*-α, *IL-6,* and caspase-8 activation pathways [29]. The results of this study are in agreement with [13], which showed that the expression levels of *TNF-α* and *IL-6* gene were induced in MCF7 cells treated with fruit extracts of *D. maritima*.

## 4. Materials and Methods

### 4.1. Plant Collection, Classification, and Extraction

The plant *D. maritima* was collected in March 2021 from Princess Tasneem bint Ghazi Technological Research Station in Al-Salt (Jordan), with coordinates: 32°05′ N 35°40′ E. The collected plant was identified by a taxonomist specialist Mr. Refad Khawaldeh at Jordan Royal Botanical Garden, where a voucher specimen (073/2021) is conserved for future reference. Then, a powder of *D. maritima* bulbs was prepared after the bulbs were cleaned with water. The grinding process was achieved manually by mortar and pestle. In the next step, the crude powdered plant (30 g) was macerated in 100 mL water for two weeks with continuous shaking at 150 rpm at RT. At the end of this period, the mixture was filtered using a clean white canvas and filter papers to yield the aqueous *D. maritima* bulb extract. Then, the aqueous part of the solution was dried out (evaporated) using a lyophilizer to obtain the desired *D. maritima* bulbs to extract in a lyophilized form. Finally, 100 mg of the *D. maritima* bulb extract was dissolved in 1 mL of DMSO to make a 100 mg/mL stock solution for the treatment experiments.

### 4.2. Identification and Characterization of D. maritima Bulb Extract by Gas Chromatography-Mass Spectrometry (GC-MS)

Gas chromatography–mass spectrometry (GC-MS) analysis of the extracted sample was performed using an Agilent 5977B GC/MSD (Agilent, Santa Clara, CA, USA). Gas Chromatograph was equipped and coupled to a mass detector Turbo Mass Gold, PerkinElmer Turbo Mass 5.1 spectrometer with an Elite-1 (100% dimethylpolysiloxane), DB-5MS, 30 m × 0.25 mm i.d., 0.25 μm film thickness of capillary column. The instrument was set to an initial temperature of 80 °C and maintained at this temperature for 1 min. At the end of this duration, the oven temperature was raised to 300 °C, at the rate of an increase of 15 °C/min, and maintained for 9 min. The injection port temperature was ensured at 290 °C, and the helium flow rate was one mL/min. The ionization voltage was 70 eV. Samples were injected in split mode as 10:1. Mass spectral scan range was set at 30–600 (*m*/*z*) [30]. 

### 4.3. Drugs Preparation

Proscillaridin A (Sigma-Aldrich, St. Louis, MO, USA) and Doxorubicin (EBEWE Pharma, Rome, Italy) were completely dissolved in DMSO and then prepared in 20 mM and 100 mM stock solutions, respectively. Then serial dilutions for both were prepared in DMEM medium and stored at −20 °C until use.

### 4.4. Cell Culture

Two human colorectal carcinoma cell lines from the American Type Culture Collection (ATCC), namely COLO-205 (ATCC^®^ CCL-222^™^) and Caco-2 (ATCC^®^ HTB-37™), were used in this study. Additionally, normal human gingival fibroblasts (hGFs) samples prepared at the Cell Therapy Center (CTC) were used to assess the cytotoxic effects of the plant extract and drugs/treatments. All cells were cultured under the same conditions in 75 cm^2^ tissue culture flasks and maintained in cell culture media (CCM) consisting of DMEM (Euroclone, Pero, Italy) high-glucose supplemented with 10% (*v*/*v*) heat-inactivated FBS, 1X penicillin/streptomycin, 1X non-essential amino acid, sodium pyruvate, and 1X L-Glutamine and incubated in a humidified incubator at 37 °C and 5% CO_2_. When the confluence of cells reached 70–80 percent, cells were passaged with 1X trypsin/EDTA (Euroclone, Italy) and centrifuged at 1400 rpm, 25 °C for five min. Different characteristics for the used cell line is listed in Table 2.

### 4.5. Cell Viability and Proliferation Assay

The antiproliferative effect and the median inhibitory concentration (IC_50_) of *D*. *maritima* bulb extract and ProA was assessed on COLO-205, Caco-2, and hGF cells using an MTT assay. In brief, cells were seeded in 96-well plates at a concentration of 8 × 10^3^ cells/well and incubated for 24 h at 37 °C. The next day, cells were treated with *D*. *maritima* bulb extract and ProA at concentrations ranging from 0.122 µg to 500 to /mL and 2.46 to 5307 µg/mL, respectively, in a quadruplicate manner. The plates were incubated at 37 °C for 48 h. COLO-205, Caco-2, and hGF cells were also treated with a 2-fold serial dilution of Doxorubicin at concentrations ranging from 0.024 to 50 µg/mL under the same conditions. DMSO was used as a drug solvent in minimal concentrations. Furthermore, each cell line used in this study was treated with 1% DMSO as a control. Additionally, untreated cells were used as a negative control. At the end of the incubation period, DMEM media were aspirated, and 10 μL of MTT dye solution (Promega, USA) was added to each well. After 4 h of incubation at 37 °C, the dye solution was removed, 100 µL of DMSO was added to each well, and the absorbance was measured using a 96-well plate reader at 560 and 750 nm wavelength. The IC_50_ values were determined using the logarithmic trend line of the cytotoxicity graph using the GraphPad PRISM^®^ 8.0 software (GraphPad Software, Inc., San Diego, CA, USA).

### 4.6. Apoptosis Assay 

Cell apoptosis was determined using the Annexin V-FITC/Propidium Iodide (PI) apoptosis detection assay Kit (Invitrogen, Waltham, MA, USA). In brief, Caco-2 and COLO-205 cells were seeded in 6-well plates at a concentration of 3 × 10^5^ cells/well, treated with different concentrations of the *D*. *maritima* bulb extract, and incubated for 24 h at 37 °C, 5% CO_2_. According to the IC_50_, Caco-2 cells were treated with different concentrations ranging from 0.5 to 2 µg/mL, COLO-205 cells were treated with different concentrations ranging from 0.5 to 3.5 µg/mL, and then both cells were incubated for 48 h at 37 °C, 5% CO_2_. The cells were harvested using 1X TrypLE Express, washed twice with PBS, and suspended in equal volumes of Annexin V and PI reagents (2 µL each) diluted in 100 µL of 1X binding buffer. All cells were incubated in a dark place at room temperature for 15–20 min. After that, cells were analyzed by flow cytometry BD FACSCanto software (BD FACSCanto, Wokingham, UK)

### 4.7. Total Reactive Oxygen Species Measurement

Seeding and treatment conditions and cell preparation were similar to apoptosis analyses. Cells were incubated with 1X ROS stain (Invitrogen, USA) (100 µL/sample) and incubated for 60 min at 37 °C, with 5% CO_2_. Then the stained cells were analyzed by flow cytometer by measuring the fluorescence emission at 520 nm.

### 4.8. Mitochondrial Membrane Potential (ΔΨm)

Mitochondrial membrane potential (ΔΨm) was assessed in colon cancer cell lines using a Tetramethylrhodamine ethyl ester (TMRE) mitochondrial membrane potential assay kit (Abcam, Boston, MA, USA). Colon cancer cells were treated as described above, and then 0.8 µL of trifluoromethoxy carbonylcyanide phenylhydrazone (FCCP) was added to the untreated cells only (control) and incubated for 15 min at 37 °C, 5% CO_2_. Treated cells were stained with TMRE by adding 0.8 µL of TMRE/well to the cell suspensions, and then cells were re-incubated for an additional 30 min. Next, cells were harvested, centrifuged, and resuspended in PBS and then analyzed by flow cytometer and measured at 549/575 nm.

### 4.9. Real-Time Quantitative Polymerase Chain Reaction (RT-qPCR) Amplification

Total RNA was extracted from cell pellets using Qiagen RNeasy Mini Kit (Qiagen, Germany), according to the manufacturer’s instructions. Complementary DNA (cDNA) was prepared using 1 µg RNA and PrimeScript RT Master Mix Kit (TaKaRa, Kusatsu, Japan) according to the manufacturer’s instructions. The cDNA synthesis reaction was performed using veriti 96 well Thermal Cycler (Thermo Fisher Scientific, Waltham, MA, USA), under the following conditions: “37 °C” (reverse-transcription) for 15 min, “85 °C” (heat inactivation for reverse-transcription) for 5 s, and “4 °C” holds. Finally, the samples were stored at −20 °C until use.

cDNA samples were diluted in 1:10 using nuclease-free water. Samples were performed in replicate. qPCR *assay* was performed *in* a 10 µL total reaction using specific primers to amplify *CASP8*, *TNF-α*, *IL-6*, and *GADPH* genes, which are responsible for the expression of caspase-8, tumor necrosis factor-alpha, interleukin-6, and glyceraldehyde-3-phosphate dehydrogenase, as the following: “95 °C” for 2 min as initial denaturation cycle, then 40 cycles of 95 °C for 15 s (denaturation), “58 °C” for 60 s (annealing), and “72 °C” for 30 s (extension). The primers sequences are listed in Table 3. The PCR data analysis was performed using the ∆∆Ct method (delta delta cycle threshold); the analysis was performed automatically according to CFX Maestro Software of Bio-Rad Company (Hercules, CA, USA). The data were normalized, across all plates, to the housekeeping gene *GAPDH*.

### 4.10. Statistical Analysis

All statistical analyses were performed using GraphPad Prism software version 8.0 (GraphPad Software, San Diego, CA, USA). The comparison between different groups of numerical variables was performed using one-way or two-way ANOVA. Homogeneity of variances was tested using Dunnett’s and Bonferroni’s multiple comparisons tests, and a *p*-value less than 0.05 (*p* < 0.05) was considered statistically significant. 

The IC_50_ values were determined by using log–probit analysis; log (concentration) plotted on the x-axis and inhibition percentage plotted on the y-axis; percent of inhibition was calculated, after correction of the absorbance (A) measurements for the background (blank) absorbance, according to the equation; Inhibition% = ((A _control_ − A_treatment_)/A_control_) × 100%.

## 5. Conclusions

In summary, *D. maritima* bulb extract and ProA showed a selective inhibitory activity on the proliferation of colorectal cancer cell lines. Such a finding was supported by the results of the apoptosis, intracellular ROS, and mitochondrial membrane potential assays. On the molecular level, the expression of pro-inflammatory genes *TNF-α* and *IL-6* and the apoptotic initiator gene *CASP8* was also confirmative for the antiproliferative effect of the *D. maritima* bulb extract. Taken together, the findings of this study elect *D. maritima* for further investigation and validation on CRC animal cancer xenograft models.

## Figures and Tables

**Figure 1 molecules-28-01215-f001:**
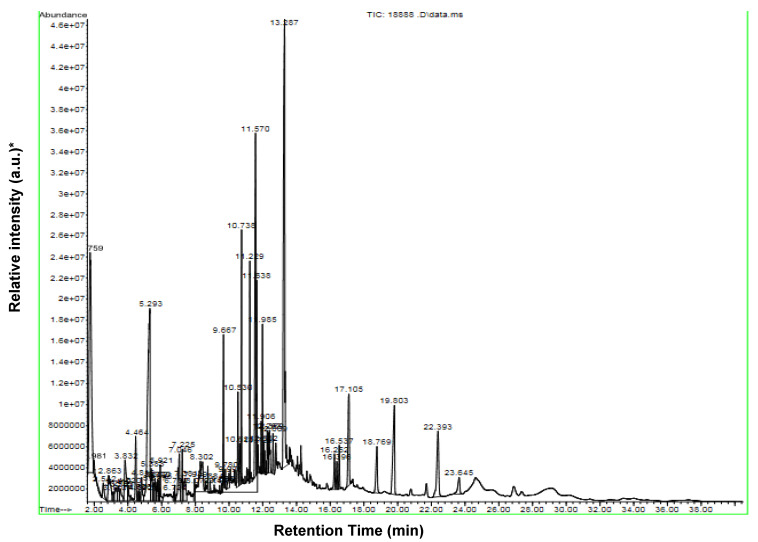
Gas chromatography and mass spectrometry chromatogram showing the major components of the aqueous bulb extract of *Drimia maritima*. * a.u.: arbitrary unit.

**Figure 2 molecules-28-01215-f002:**
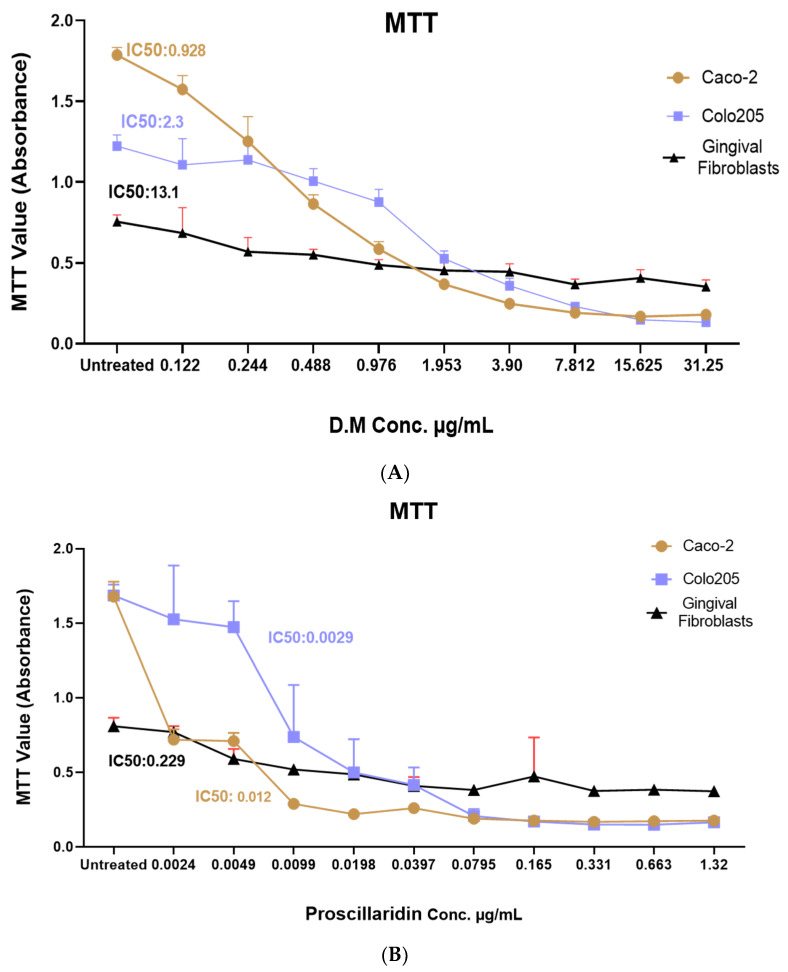
Cell viability MTT assay of Caco-2 and COLO-205 colon cancer cells and hGFs after treatment with the following: (**A**) increasing concentration of aqueous *Drimia maritima* bulb extract (0.122–31.25 µg/mL); (**B**) Proscillaridin (0.0024–1.32 μg/mL); (**C**) Doxorubicin (from 50 to 0.195 µg/mL). The IC_50_ values were calculated using log-probit analysis. Each value is presented as the mean ± SD of an average of three independent experiments.

**Figure 3 molecules-28-01215-f003:**
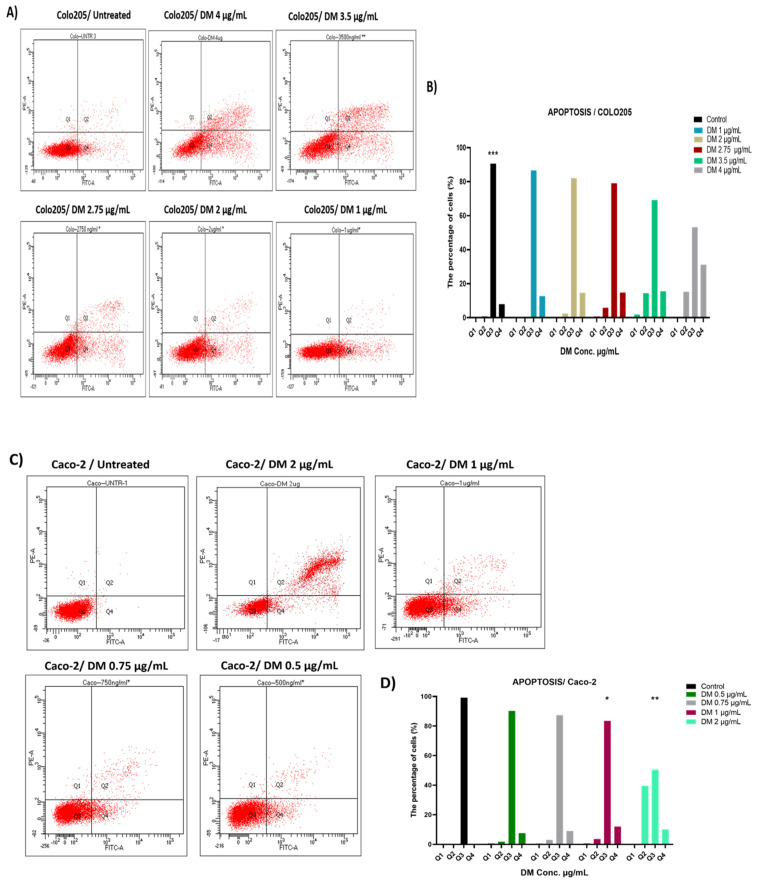
Flow cytometry analysis of D. maritima (DM) bulb extract effect on COLO-205 and Caco-2 cells. (**A**) Representative flow cytometer plots are presented for the untreated group (control) and DM-treated groups (4 μg/Ml, 3.5 μg/mL, 2.75 μg/mL, 2 μg/mL, 1 μg/mL and 0.5 μg/mL) in COLO205 cells. (**B**) The bar graph represents the percentage of early and late apoptotic cells detected by flow cytometer from three separate experiments (mean ± SD, n = 3). (**C**) Representative flow cytometery plots are presented for the control group (Untreated) and DM treated groups (2 μg/mL, 1 μg/mL, 0.75 μg/mL, and 0.5 μg/mL) in Caco-2 cells. (**D**) The bar graph represents the percentage of early and late apoptotic cells detected by flow cytometer from three different individual experiments (mean ± SD, n = 3). ** Significant differences were observed between the DM-treated (2 μg/mL and 1 μg/mL) and untreated control group (*p*-values *** < 0.001, ** < 0.01 and * < 0.05).

**Figure 4 molecules-28-01215-f004:**
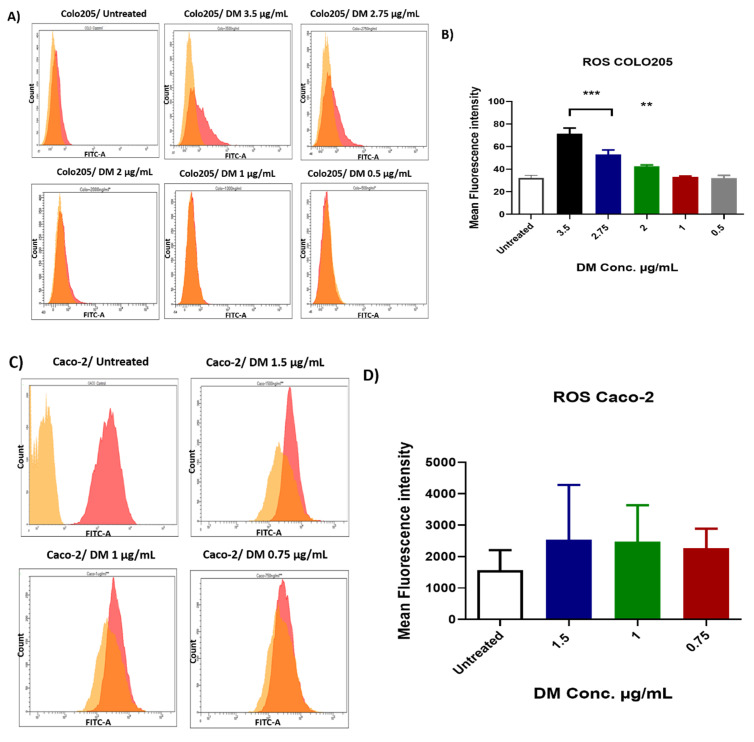
Production of reactive oxygen species (ROS) in COLO-205 and Caco-2 cells incubated for 48 h with *D. maritima* (DM) bulb extract: (**A**,**B**) Histograms and bar graphs of ROS production in COLO-205 cells were obtained by flow cytometer in the FITC channel in different groups. The bar graph shows a remarkable increase in the level of intracellular ROS in the treated group. (**C**,**D**) Representative histograms and bar graphs from Caco-2 cells treated with indicated concentrations of *D. maritima* extract on intracellular ROS levels production in comparison to control (untreated cells) detected by flow cytometer from three separate experiments (mean ± SD, n = 3). *p*-values *** < 0.001, ** < 0.01.

**Figure 5 molecules-28-01215-f005:**
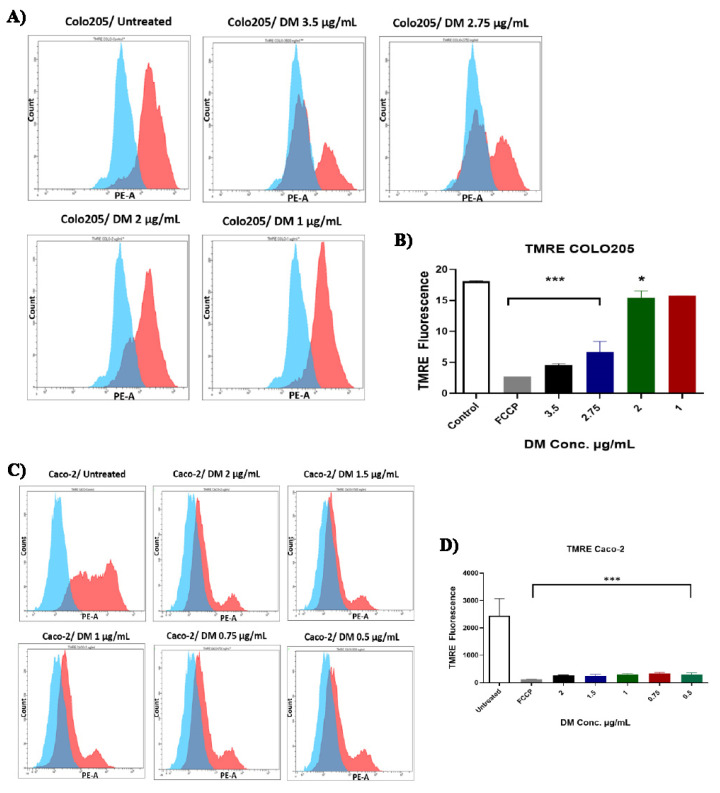
Assessment of the mitochondrial membrane potential (ΔΨm) in COLO-205 and Caco-2 cells after treatment with *D. maritima* (DM) bulb extract for 48 h: (**A**,**B**) Histograms and bar graphs show changes in COLO-205 cells ΔΨm response to various concentrations of DM extract (3.5 μg/mL, 2.75 μg/mL, 2 μg/mL, 1 μg/mL, and 0.5 μg/mL) in comparison to untreated cells. (**C**,**D**) Histogram and bar graphs show loss of ΔΨm following exposure to different concentrations of DM extract (2 μg/mL, 1.5 μg/mL, 1 μg/mL, 0.75 μg/mL, and 0.5 μg/mL) compared to control (untreated cells). All data are expressed as mean  ±  SD of three separate experiments. All data are expressed as mean ± SD of three separate experiments. (*p*-values *** < 0.001 ** < 0.01, and * < 0.05).

**Figure 6 molecules-28-01215-f006:**
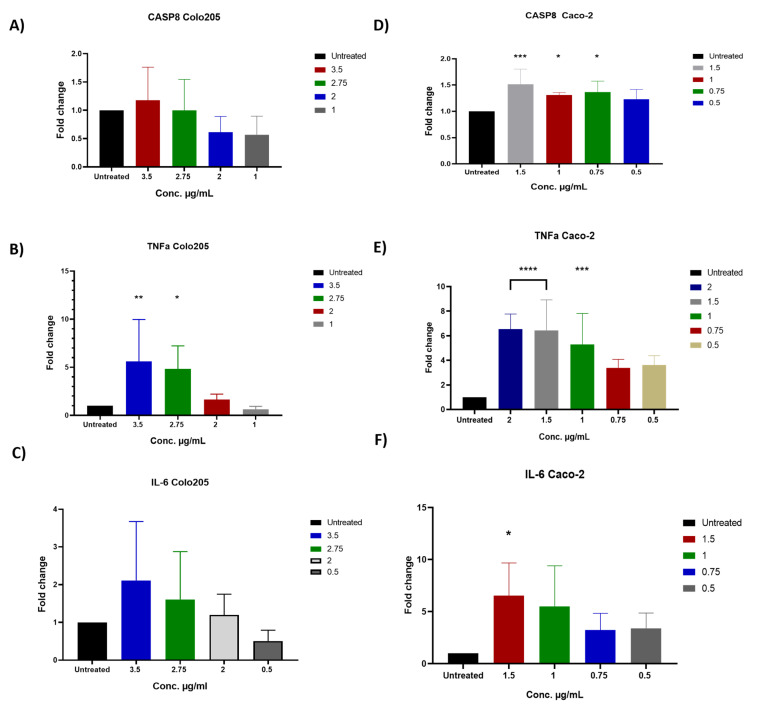
Gene expression analysis of apoptotic initiator gene *CASP8* and two inflammatory cytokine genes *TNF-α* and *IL-6* in COLO-205 and Caco-2 cells after treatment with various concentrations of *D. maritima* (DM) bulb extract for 48 h as examined by quantitative RT–PCR: (**A**,**D**) The graphs reveal fold changes in the expression of *CASP8* in COLO-205 and Caco-2 cells, respectively. (**B**,**E**) *TNF-α* gene in COLO-205 and Caco-2 cells at various extract concentrations (2.0, 1.5, 1.0, 0.75, and 0.5 μg/mL). (**C**,**F**) *IL-6* in COLO-205 and Caco-2 cells treated with different extract concentrations (1.5, 1.0, 0.75, and 0.5 μg/mL). Fold change was calculated using the ΔΔCt method. Untreated cells were used as the control; therefore, the fold change for untreated cells is 1 in all plots. *GAPDH* gene served as an internal reference gene. The error bars indicate the standard deviation from triplicate experiments (* *p* < 0.05; ** *p* < 0.01; *** *p* < 0.001; **** *p* < 0.0001).

**Table 1 molecules-28-01215-t001:** Gas chromatography and mass spectrometry data of chemical components of *Drimia maritima* bulb extract.

No.	Component	Cas #	Content (%)
** 1 **	2,3-Butanediol	513-85-9	10.454
** 2 **	Furancarboxaldehyde	98-01-1	0.28
** 3 **	Isobutanoic acid	79-31-2	0.544
** 4 **	2-Furancarboxaldehyde, 5-methyl	620-02-0	1.389
** 5 **	Erythritol	149-32-6	0.267
** 6 **	Oxetane, 3,3-dimethyl	6921-35-3	0.626
** 7 **	Limonene	138-86-3	0.449
** 8 **	Methyl-3-furanthiol	28588-74-1	0.523
** 9 **	2H-Pyrazole-3-carbohydrazide	9-64-26275	1.82
** 10 **	Butanedioic acid, monomethyl ester	3878-55-5	0.379
** 11 **	4H-Pyran-4-one, 2,3-dihydro-3,5-dihydroxy-6-methyl	28564-83-2	1.333
** 12 **	(E,5S)-3,5-dimethylhept-3-en-1-yne	997029-22-6	0.219
** 13 **	2-Furancarboxaldehyde, 5-(chloromethyl)(	1623-88-7	0.142
** 14 **	4H-Pyran-4-one, 3,5-dihydroxy-2-methyl	1073-96-7	0.666
** 15 **	5-Hydroxymethylfurfural	67-47-0	9.857
** 16 **	α-D-Glucopyranoside, O-α-d-glucopyranosyl-(1.fwdarw.3)-β-d-fructofuranosyl	597-12-6	0.843
** 17 **	α-D-Glucopyranoside, O-α-d-glucopyranosyl-(1.fwdarw.3)-β-d-fructofuranosyl	597-12-6	0.482
** 18 **	3-Methoxybenzyl alcohol	6971-51-3	0.538
** 19 **	5-Acetoxymethyl-2-furaldehyde	10551-58-3	0.449
** 20 **	5-Hydroxymethylfurfural	67-47-0	0.335
** 21 **	Glutaric acid, 2-naphthyl tridecyl ester	998725-55-8	0.56
** 22 **	8-Oxabicyclo[5.1.0]oct-5-en-2-ol, 1,4,4-trimethyl	58795-43-0	0.14
** 23 **	Cycloheptasiloxane, tetradecamethyl	107-50-6	0.16
** 24 **	Ethyl hydrogen succinate	1070-34-4	1.679
** 25 **	Phenol, 2,4-bis(1,1-dimethylethyl)	96-76-4	1.029
** 26 **	β-D-Glucopyranose, 1,6-anhydro	498-07-7	1.072
** 27 **	α-D-Glucopyranoside, O-α-d-glucopyranosyl-(1.fwdarw.3)-β-d-fructofuranosyl	597-12-6	0.217
** 28 **	6Methoxy-2-amido-5,6-dihydrothiazolo[2,3-c]-1,2,4-triazole	997204-67-5	3.361
** 29 **	Myristic acid	544-63-8	0.155
** 30 **	Ethyl hydrogen succinate	1070-34-4	0.339
** 31 **	Hexadecanoic acid, 1-(hydroxymethyl)-1,2-ethanediyl ester	761-35-3	0.067
** 32 **	Palmitoleic acid	373-49-9	0.063
** 33 **	n-Hexadecanoic acid	57-10-3	0.105
** 34 **	Hexadecanoic acid, ethyl ester	628-97-7	8.057
** 35 **	Heptadecanoic acid	506-12-7	0.159
** 36 **	Octadecanoic acid	57-11-4	0.417
** 37 **	Octadecenoic acid (Z) 9	112-80-1	1.578
** 38 **	Octadecanoic acid	57-11-4	0.73
** 39 **	Hexadecanamide	629-54-9	2.309
** 40 **	Stearic acid, 2-hydroxy-1-methylpropyl ester	14251-39-9	1.497
** 41 **	9-Octadecenamide, (Z)	301-02-0	8.142
** 42 **	Octadecanamide	124-26-5	1.513
** 43 **	(Z)-(S)-Octadec-9-en-11-olide	997490-85-5	3.409
** 44 **	n-Propyl 9-octadecenoate	997641-34-3	0.439
** 45 **	Stearic acid, 2-hydroxy-1-methylpropyl ester	14251-39-9	1.569
** 46 **	Bis(2-ethylhexyl) phthalate	117-81-7	0.499
** 47 **	Elaidamide	301-02-0	0.475
** 48 **	Ethyl iso-allocholate	112-84-5	1.244
** 49 **	13-Docosenamide, (Z)	112-84-5	0.545
** 50 **	Campesterol	474-62-4	13.874
** 51 **	6,12-dimethoxy-8-methyl-5,8,9,13-tetrahydro-7H-cyclohept[b]anthracene-5,13-dione	997713-65-8	0.723
** 52 **	Stigmasterol	83-48-7	0.491
** 53 **	β-Sitosterol	83-46-5	0.735
** 54 **	Bufa-20,22-dienolide, 14-hydroxy-3-oxo-, (5β)	4029-65-6	2.642
** 55 **	Proscillaridin	466-06-8	4.956
** 56 **	4-(2,4-Dimethyl-phenyl)-1,7-dimethyl-4-azatricyclo[5.2.1.0(2,6)]decane-3,5,8-trione	997597-94-2	2.78
** 57 **	Bufa-20,22-dienolide, 3-(acetyloxy)-14,15-epoxy-5-hydroxy-, (3β,5β,15β)	4029-68-9	0.766

**Table 2 molecules-28-01215-t002:** Type of cells used in the study.

Cell Line	Organism	Tissue	Morphology	Culture Properties	Mutant Gene	Chemoresistance
**COLO-205** **(ATCC^®^ CCL-222^™^) ***	*Homo sapiens*, human	Colon	Epithelial	Mixed; adherent and suspension	APC, BRAF SMAD4, TP53	Cisplatin
**Caco-2** **(ATCC^®^ HTB-37^™^) ***	*Homo sapiens*, human	Colon	Epithelial	Adherent	APC, SMAD4 TP53	5-fluorouracil

*; https://www.atcc.org (accessed on 20 December 2022).

**Table 3 molecules-28-01215-t003:** List of primers for qPCR amplification.

Gene	Forward Primer Sequence (5′-3′)	Reverse Primer Sequence (5′-3′)
*CASP8*	CTG CTG GGG ATG GCC ACT GTG	TCG CCT CGA GGA CAT CGC TCT C
*TNF-a*	GTC AAC CTC CTC TCT GCC AT	CCA AAG TAG ACC TGC CCA GA
*IL-6*	TTC CAA AGA TGT AGC CGC CC	ACC AGG CAA GTC TCC TCA TT
*GAPDH*	CCT GTT CGA CAG TCA GCC G	CGA CCA AAT CCG TTG ACT CC

## Data Availability

All data are available upon request from the corresponding author.

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
