# Peer review of "Phytochemical Analysis and Anticancer Properties of Drimia maritima Bulb Extracts on Colorectal Cancer Cells"

_molecules, 2023, doi:10.3390/molecules28031215_

Round 1
Reviewer 1 Report
Title: Molecular Evaluation of Anticancer Effect of Drimia maritima Bulb Extracts on Colorectal Cancer Cells
1. Please provide a voucher specimen and the herbarium where the sample was deposited.
2. Please provide the SI (selectivity index) and discuss. This will provide you some information about how selective your extract is.
3. Please change ml to mL for in all figs.
4. Line 281-282: Do you have the same SD? How is that possible?
5. Fig 4, Your flow cytometer's setting seemed problematic. Please provide the good one with clear G1, S and G2 area. These narrow peaks will surely affect your data quantification.
6. Line 310 and 313 Do you have the same SD?
7. Line 313 22.58 without SD?
8. Line 320 and 323 Do you have the same SD? I am now really concerned with the accuracy of the data. Also line 360 and 367. Please check the whole manuscript
9. I suggest the authors reorder the dose of the extract, it should start with low to high concentration, not high to low.
10. Why did you use Caco-2 and COLO-205? And since the extract showed different effects on these two cells, please add in the discussion, why?
11. You measured Cas8, why not Cas9 which directly correlate with membrane potential.
12. 143 refs for one article is extremely high, please trim it to not more than 60.
13. Please rewrite conclusion, it is like an abstract.
Author Response
NA
Please see the attachment

Reviewer 2 Report
The article titled: Molecular Evaluation of Anticancer Effect of Drimia maritima Bulb Extracts on Colorectal Cancer Cells” may be a valuable manuscript on D. maritima biological potential and chemical characterization. However, before publication, it demands some essential amendments, and some parts of this article should be improved. Below I present my comments.
Title:
- Only three genes were examined in the study (only at the mRNA level), so it's hard to discuss the molecular aspect of work. (And no significant changes in their expression are observed).
- I would change the article title:" Phytochemical analysis and anticancer properties of D.maritima bulb aqueous extract on colon cancer cell lines". something like that..
Abstract:
- Needs to be changed. Numerous factual errors and inconsistencies with the obtained results are observed.
- 32 line….induce intracellular ROS level -- what induce ? increased/decreased? Need to be specified.
- 33 line … What kind of response ??
- 37 line- CASP8 expression was not increase at tested extract concentration on COLO205 cell line.
Keywords:
- Needs to be change : colorectal ?? … cancer; anticancer ??.. properties .. These are adjectives , should be nouns.
Introduction:
- Enormous grammatical English errors
- Two paragraphs should be deleted or shortened. It is an original work, not a review. It contains basic information.. not relevant to the article.
- 43 line ... "In the multicellular human body", Can the human body be unicellular ?
-74-77 lines - „ immense suffering in diseased humans” gramma- I find it difficult to understand.
- 80 line , one of the most common cancer types in Jordan as well as in the world.. can be removed,
- The study aimed to collect D.maritima bulbs ? Whether the assessment of the anticancer properties of extracts.
Materials and methods:
The manuscript needs many corrections. I recommend making just a table with the used extract concentrations on cell lines. and delete this information (about extract concentration) everywhere to make a manuscript clearer.
Drug preparation:
- Source of ProA and Doxorubicin… in what they have been dissolved -DMSO? DMSO stock solution ??
- ProA- expand the abbreviation
- Under what conditions the doxorubicin was stored. Doxorubicin should be stored in the fridge to protect from light and its degradation in higher temperatures. How long they were stored in dilute solutions before being used? Cell culture:
Cell culture:
- Why did you choose these two cell lines ?? describe their: differences. p53 status ? mutations ?? chemoresistance / chemosensitive ??
- If the cell lines were purchased from ATCC why they are not growing on their recommended medium such as EMEM for Caco-2 and RPMI1680 for COLO-205 but on DMEM?
- DMEM, Trypsin/EDTA – source
- 125 line –drugs/treatments?
Cell viability and proliferation assay:
- Did the authors pay attention to the doubling time of the cell lines used? whether the number 8x103 is sufficient to reach sufficient cell confluence after 24h to add test compounds.?
Doublin time for COLO-205 – 25 hours, for Caco-2 - 3-4 days (ATCC); ~80 hours (DSMZ); ~32 hours (PBCF); ~60-70 hours (CLS).
- ProA and Doxorubicin concentrations (µg / ml or mM) should be unified
- 144 line- MTT solution – source and concentration ?
- Why the absorbance was measured at 750 nm wavelength?
- Did you investigate the effect of the extract alone (without cells) on the ability to reduce MTT salt? It sometimes happens with the descent natural products possess such abilities.
- did you check plates under a microscope before adding mtt solution?
Apoptosis assay:
- What plates were you using? 6 or 12 wells?
- 154 line …. Why now are you performing test after 24h and not like the cytotoxic assay after 48h? You present ic50 for 48h, not 24h.
- 157 lines …. Now 48h ?? so how long do you incubate cells with extract?
- Flow cytometry - on what equipment?
Cell cycle analysis:
- 163 line – “sterile” .. it's obvious, so I recommend to delete.
- 168 line – choose rcf/rpm or G and unify in all metods
- PI and 200xRNAse – source
Total reactive oxygen species measurement:
- 177 line- ROS stain - source
Real-Time quantitative polymerase chain reaction (RT-qPCR) amplification:
- 195 line…. “37 °C”, “85°C”,
- 196, 202, 203 lines .. the same …
Statistical analysis:
- Why did you choose one-way or two-way ANOVA. T-student or Mann-Whitney test should better in this case.
Results:
Drimia maritima bulb extract and ProA can selectively inhibit the proliferation of COLO-205 and Caco-2 cancer cells.
- “Drimia maritima bulb extract and ProA can selectively inhibit the proliferation of COLO-205 and Caco-2 cancer cells.” – here mtt assay measures the viability of cells and also the cytotoxicity of tested extract and ProA not their ability to inhibition of proliferation.
- Why you used Doxorubicin as positive control ? Is it used normally to treat colon cancer?
- Can the concentration of ProA in nM be compared to its concentration in the tested extract? It will be useful if you change the unit of used ProA to ug/ml.
- How long do you incubate mtt solution with fibroblast.? ? the mean absorbance below 1 represents that these cells possess slow metabolism. Did you check the plate under the microscope how mtt reacted?
- Table 2. -many punctuation errors in compounds name
- Figure 2. The name of cell lines should be the same in all figure.. Caco-2 or CACO , colo205 or COLO,
- The same “0” or “untreated”
- 2C – where is positive control to fibroblast, DOX abbreviation, the concentration should increase on the x-axis not decrease.
- Where is statistic. (STARS ?)
D. maritima bulb extract can induce early and late apoptosis in colon cancer cell lines.
- why you did not divide apoptotic cells into early and late apoptotic just % of apoptotic. The early stage of apoptosis is reversible thus it is necessary to divide the cells into early and late-apoptotic.
- “In COLO-205 cell culture, there was a statistically significant increase in the percentage of apoptotic cells (9.1%±2.299, 9%±2.299, 9.8%±2.299, 12.52%±2.299, 15.46%±2.299, and 22.8%±2.299) according to the D. maritima bulb extract concentration (0 "control", 0.5, 1, 2, 2.75, and 3.5 μg/mL), respectively (Figure 3A).” 3.5ug/ml concentration is much higher than IC50 and % of apoptotic cells is only 12 % higher than the control. (what the % of it is early apoptotic cells)
- “ Similarly, after 48 h of treatment with different concentrations (0 "control", 0.5, 0.75, 1, and 2 μg/mL) of D. maritima bulb extract, a statistically significant increase in the percentage of cell apoptosis was observed (1.4% ±1.197 "control", 1.197±3%, 1.197±4.16%, 1.197±6%, and 6.23%±1.197, respectively) in Caco-2 cell culture”. IC50 is almost 1ug/ml the % of apoptotic cells is even lower than the control. In higher (2-times higher) extract concentration the % of apoptotic cells increases to 6%. I understand that these results are statistically significant but did they biologically significantly?
Effect of D.maritima bulb extract on cell growth and cell cycle of colon cancer cell lines
- Why you divided cells on G0/G1 and S/G2 phases - it is not typical. It should be divided into Subg1 (optional), G1/G0, S, and G2/M.
- In normal cell cycle analysis, the % of cells in G0/G1 is higher than the sum of the % of s and g2/m phases. –CONTROL
- “For untreated (control) COLO-205 cells, the percentages of cells in G0/G1 and S/G2 phases were 13.56%, and 24.36%, respectively”. The sum of % G0/G1 and G2/M phase is 37.92 % where are 62,05 % of cells? on which phases of the cell cycle.
- Similar for CACO cell line… See above
- In the tested concentrations, the extract did not induce changes in the cell cycle of the tested cells
D. maritima bulb extract induces the production of ROS in colon cancer cells.
- 339 line … “p-value >0.001”
- 368 line – (Figure 5) should be Figure 6.
- Figure 6D – "the stars" under other bars not under control - Statistical significance.
- 418 line. Figure 6 – change numeration -should be 7
- Figure 7 ABC – C Why you exclude 1ug/ml and added 0.5ug/m?
- Figure 7DEF – D ,F– where is 2ug/ml;
Discussion
- The author repeats the introduction 428-452 - (introduction). It does not compare the obtained results with other similar articles (For example, IC50 or phytochemical analysis). Apparently, the studied extract has already been tested in relation to other cancers, including colon cancer (but on different cell lines).
- 483-492lines - This is methodology not discussion.
- The discussion requires a lot of changes . These are the results and methodology paraphrased for the second time.
- Too many references for an original article !!
- It is not a review.
- References should be numbered according to the order in which they appear in the text, not alphabetically.
Conclusion
- 573 line –apoptosis. statistically significant but not biological …
- 575 line - ROS production was noted only on COLO 205 cell line not both… And the extract did not inhibit cell cycle
- 579- “in this study in both cell lines,…. it can upregulate the expression of inflammatory cytokine genes TNF-α and IL6 and the apoptotic initiator gene CASP8” not true …. Figure 7A and C.
- The Discussion and conclusion should be rewritten
Author Response
NA
Please see the attachment

Round 2
Reviewer 1 Report
I asked “10. Why did you use Caco-2 and COLO-205? And since the extract showed different effects on these two cells, please add in the discussion, why?”
The authors said that “Just to show that 2 different type of colon cancer with different characteristic have responded to our extract”.
This cannot be considered a scientific response. Why did you choose these particular cells? You must provide a molecular explanation and clarify why these two cell types respond differently to your extract.
Author Response
Thanks much for your comments. It has been taken in consideration and the following the following paragraph has been added to the discussion part.
The Colo-205 represent a n metastatic colorectal cancer as this cell was isolated from ascitic fluid of a 70-year-old Caucasian male with carcinoma of the colon. The patient had been treated with 5-fluorouracil for 4-6 weeks before removal of the fluid specimen. While Caco-2 cells represent a relatively sensitive cells toward the 5-luorouracil. Interestingly, the extract used in this study show a relatively higher activity toward the colorectal Caco-2 cells. Interestingly the apoptosis assay also have shown less apoptosis percentage that those seen in Caco-2 cells case.

Reviewer 2 Report
Dear authors
Even though you made some changes to the manuscript, you did not answer my questions and did not try to correct the manuscript according to a few recommendations. The manuscript still has many errors, which should be excluded or fixed before publishing.

Author Response
The abstract should be a total of about 200 words maximum. Your abstract is almost twice as long.
Thank you for the comments. Ut has been shortened accordingly.
- The intracellular ROS levels in colon (cancer cell line ???) was also induced.
This sentence was deleted and corrected accordingly.
Keywords: Where are the keywords?
Key words was added.
Introduction: - Two paragraphs should be deleted or shortened. It is an original work, not a review. It contains basic information not relevant to the article. Was corrected accordingly – what was deleted and shortened?
The introduction was shortened accordingly. The whole unnecessary paragraph was deleted.
- 43 line ... "In the multicellular human body", Can the human body be unicellular ? just human body Was corrected accordingly and the following sentence was added In the incredibly complex human body.
-74-77 lines - „ immense suffering in diseased humans” gramma- I find it difficult to understand. Was corrected accordingly and the following sentence was added as a source for safe and effective anticancer treatment.
- 80 line , one of the most common cancer types in Jordan as well as in the world.. can be removed, Well done.
- The study aimed to collect D.maritima bulbs ? Whether the assessment of the anticancer properties of extracts. Was corrected accordingly The introduction has not been corrected sufficiently.. Methods: Cell cycle analysis: It was not corrected. it was just deleted. Why? - 163 line – “sterile” .. it's obvious, so I recommend to delete. Was corrected accordingly
- 168 line – choose rcf/rpm or G and unify in all metods Was corrected accordilgly - PI and 200xRNAse – source Was corrected accordingly
- maritima bulb extract can induce early and late apoptosis in colon cancer cell lines.
- why you did not divide apoptotic cells into early and late apoptotic just % of apoptotic?. The early stage of apoptosis is reversible thus it is necessary to divide the cells into early and late-apoptotic. - “In COLO-205 cell culture, there was a statistically significant increase in the percentage of apoptotic cells (9.1%±2.299, 9%±2.299, 9.8%±2.299, 12.52%±2.299, 15.46%±2.299, and 22.8%±2.299) according to the D. maritima bulb extract concentration (0 "control", 0.5, 1, 2, 2.75, and 3.5 μg/mL), respectively (Figure 3A).” 3.5ug/ml concentration is much higher than IC50 and % of apoptotic cells is only 12 % higher than the control. (what the % of it is early apoptotic cells) - “ Similarly, after 48 h of treatment with different concentrations (0 "control", 0.5, 0.75, 1, and 2 μg/mL) of D. maritima bulb extract, a statistically significant increase in the percentage of cell apoptosis was observed (1.4% ±1.197 "control", 1.197±3%, 1.197±4.16%, 1.197±6%, and 6.23%±1.197, respectively) in Caco-2 cell culture”. IC50 is almost 1ug/ml the % of apoptotic cells is even lower than the control. In higher (2-times higher) extract concentration the % of apoptotic cells increases to 6%. I understand that these results are statistically significant but did they biologically significantly?
The experiments were analyzed again and re-constructed in four different quartiles. Please see the new apoptotic figures.
Effect of D.maritima bulb extract on cell growth and cell cycle of colon cancer cell lines - Why you divided cells on G0/G1 and S/G2 phases - it is not typical. It should be divided into Subg1 (optional), G1/G0, S, and G2/M. - In normal cell cycle analysis, the % of cells in G0/G1 is higher than the sum of the % of s and g2/m phases. –CONTROL - “For untreated (control) COLO-205 cells, the percentages of cells in G0/G1 and S/G2 phases were 13.56%, and 24.36%, respectively”. The sum of % G0/G1 and G2/M phase is 37.92 % where are 62,05 % of cells? on which phases of the cell cycle. - Similar for CACO cell line... See above - In the tested concentrations, the extract did not induce changes in the cell cycle of the tested cells. Where are the responses??
Regarding to the Cell Cycle data, we had a problems when we analyzing the data and we thought it gowing to be enough in the current situation to be representative in the manuscript however, we think now of removing it from the whole manuscript and it will not affect the quality of the work. we think keeping it in its current form will not be very suitable for your journal so have deleted the whole cell cycle results, the figures and anything related to cell cycle from the whole manuscript.
Discussion: It does not compare the obtained results with other similar articles (For example, IC50 or phytochemical analysis). Apparently, the studied extract has already been tested in relation to other cancers, including colon cancer (but on different cell lines). Compared to the earlier version of the discussion, some information has only been removed. The information I sought, e.g. comparing the ic50 values for the tested extract on other cell lines, has not been added.
The following was added to the discussion
Interestingly, the obtained IC50 values in this study is relatively lower than the previous reported IC50 against deferent breast cancer cell lines. For Example, Hamzeloo-Moghadam reported that the IC50 values for the non-hemolytic ethanol extract of U. maritima against breast cancer MCF7 cells was 11.01 µg/ml. Whereas, IC50 values for the non-hemolytic acetone extract of U. maritima against breast cancer MCF7 cells was 6.01 µg/ml. While, Doxorubicin drug (positive control) exhibited cytotoxic effect with higher IC50 value (52.35 µg/ml) (20). Another example is the study that conducted by Obeidat and Sharan and found that The effective doses of D. maritima that inhibited 50% of growth (IC50) of MCF-7 and MDA-MB-468 cells were 20.48±1.17 µg/mL and 25.74±2.05 µg/mL, respectively. Interestingly, Obeidat and Sharab study revealed that D. maritima display significantly lower cytotoxicity against AGO1522, a normal human fibroblast cell, with the IC50 values of 43.5±1.73 µg/mL (21).
